# Knowledge Levels and Learning Needs in Dysphagia Management: Perspectives from Professional and Non-Professional Stakeholders in Five European Countries

**DOI:** 10.3390/healthcare13233140

**Published:** 2025-12-02

**Authors:** Sara Remón, Ana Ferrer-Mairal, Vijolė Bradauskienė, Ana Cristina Cortés, Teresa Sanclemente

**Affiliations:** 1Departamento de Producción Animal y Ciencia de los Alimentos, Instituto Agroalimentario de Aragón (IA2), Universidad de Zaragoza-CITA, M Servet 177, 50013 Zaragoza, Spain; remon@unizar.es (S.R.); ferrerma@unizar.es (A.F.-M.); 2Facultad de Ciencias de la Salud y del Deporte, Universidad de Zaragoza, Pl. Universidad 3, 22002 Huesca, Spain; 3Food Technologies and Nutrition Department, Klaipėdos Valstybinė Kolegija/Higher Education Institution, Bijūnų Str. 10, LT-91223 Klaipėda, Lithuania; v.bradauskiene@kvk.lt; 4CADIS Huesca, Coordinadora de Asociaciones de Discapacidad, 22002 Huesca, Spain; anacortes@cadishuesca.es

**Keywords:** dysphagia management, learning, caregivers, healthcare professionals, cross-sectional study

## Abstract

**Background/Objectives:** Dysphagia represents a significant global health concern with particularly high prevalence in specific clinical conditions, yet educational gaps persist among healthcare professionals and caregivers. This observational, cross-sectional quantitative study aimed to provide a comprehensive overview of the current self-perceived knowledge and learning needs among stakeholders involved in dysphagia management. **Methods:** An international online survey was conducted in five European countries (Greece, Italy, Lithuania, Spain, and Turkey) with 463 participants: 297 professionals (healthcare and non-health specialists, educators, students) and 166 non-professionals (patients, family members, caregivers, interested individuals). Two structured questionnaires explored self-perceived knowledge, learning needs, relevancy of thematic areas, and preferred learning methods. Chi-square and Fisher’s exact tests were used for statistical comparisons. **Results:** Professionals reported higher self-perceived knowledge than non-professionals (*p* < 0.001), yet both groups expressed comparable needs for further education. Priority learning areas varied by respondent profile: “Identification & Treatment” was prioritized by both speech-language pathologists and healthcare specialists, as well as by non-professionals, while dietitian-nutritionists focused on “Diet & Nutrition” and “Food Preparation”. Short-duration courses and visual, hands-on learning tools were preferred across groups. **Conclusions:** This study highlights a broad demand for dysphagia education among professionals and non-professionals. Tailored, technology-enhanced learning programs could bridge existing knowledge gaps, strengthen multidisciplinary collaboration, and support better daily management of dysphagia.

## 1. Introduction

Dysphagia is defined as difficulty or inability to perform normal swallowing, that is, the safe transport of food and liquids from the mouth to the stomach. Dysphagia is commonly observed worldwide, and its trend has been increasing in recent years [1]. Its prevalence varies considerably depending on the population studied. In patients with neurological conditions, the global prevalence reaches 50.4% in Parkinson’s disease [2] and 44.8% in multiple sclerosis [3]. Following acute stroke, the pooled incidence of poststroke dysphagia is 42% [4]. In head and neck cancer patients, dysphagia is present in up to 28% at diagnosis, but can increase to 45–75% due to treatment [5]. In healthcare settings, prevalence ranges from 36.5% in hospitals [6] to 56.1% in residential aged care facilities [7]. Dysphagia is associated with serious complications such as malnutrition, dehydration, and aspiration pneumonia, as well as psychological and social consequences that negatively affect quality of life [8,9].

Proper management of dysphagia involves early detection, active treatment, and compensatory strategies. The latter includes adapting food texture and drink viscosity to individual needs and controlling posture during swallowing [10]. These interventions require a multidisciplinary team, including speech-language pathologists (SLPs), dietitians-nutritionists (D-Ns), nurses, physical and occupational therapists, and auxiliary care staff [4,9]. However, in many clinical settings, limited resources restrict access to comprehensive dysphagia care teams, and individual healthcare providers are often required to take responsibility for all aspects of this management. In this sense, several studies among healthcare professionals indicate a clear demand for improving their competencies and skills in this field. Specifically, research among SLPs in the US [11,12], Japan [13], Norway [14], and Australia [15], as well as among nurses in Norway [16], Portugal [17], and Japan [13], consistently points to gaps in dysphagia-related expertise. Similar findings have been reported for broader groups of healthcare professionals in Iran [18] and Spain [4], and for nursing home staff in the UK [19], care homes in Norway [20], intensive care units in the Netherlands [21], and in multi-country studies covering 26 nations [22]. Importantly, previous studies indicate that more than 40% of professionals working in residential centers lack the necessary knowledge for adequate dysphagia management [23,24]. As a result, there is consensus in the literature on the urgent need for enhanced training for both qualified and unqualified care staff [19,20], supported by structured, high-quality continuing professional development programs [4]. The recent TEAMS international survey (877 health professionals from 54 countries) confirmed inadequate knowledge and inconsistent use of validated nutrition screening tools, highlighting the urgent need for targeted education programs [25]. Furthermore, because compensatory management of dysphagia also involves non-healthcare professionals—including food technologists [26], food service employees [27], and healthcare managers [28]—educational initiatives should extend beyond clinical staff to these groups as well.

On the other hand, caregivers and family members play a crucial role in early recognition and daily management of dysphagia [29,30], but recent studies highlight the challenges they face due to limited knowledge, lack of training, and emotional burden [31,32]. In addition, recent research has shown that dysphagia substantially reduces quality of life for patients, their supporters, and healthcare professionals [33]. This wide-ranging impact underscores the importance of including all perspectives in learning strategies. Furthermore, patients themselves must be involved in these programs, as they require clear and accessible information about their condition [34]. While few studies have explored the perceptions of individuals with dysphagia, existing research supports the role of education in improving management and outcomes [35,36]. Despite its relevance, public awareness of dysphagia remains low. A recent survey of US adults revealed limited understanding of the condition, underscoring the need to extend educational efforts beyond clinical settings [37].

Recent studies have emphasized the need to strengthen knowledge translation and educational strategies in healthcare, both within dysphagia management and in broader clinical contexts. Research has shown that improving caregiver training and communication tools can enhance safety and quality of care for patients with dysphagia [38,39], while other studies highlight the importance of structured learning approaches and skills development among health professionals [40,41,42]. However, designing effective educational interventions requires first understanding the current knowledge levels and learning needs across all stakeholder groups—a gap that remains poorly characterized in the literature. Building on this gap, the present study aimed to provide a comprehensive overview of the current self-perceived knowledge and learning needs among stakeholders involved in dysphagia management. An international survey was conducted among professionals and non-professionals involved in dysphagia management to explore: (1) self-perceived knowledge levels and identified educational and training needs, (2) priority topics of interest for each stakeholder group, and (3) preferred learning formats and methodologies. Understanding these perspectives will support the development of targeted educational interventions for all those involved in dysphagia care: professionals, individuals with dysphagia, and their caregivers.

## 2. Materials and Methods

### 2.1. Study Design and Participants

The research formed part of the Erasmus+ project INDEED “Innovative tools for Diets oriented to Education and Health Improvement in Dysphagia condition (KA204-083288)”, which involved five countries: Greece, Italy, Lithuania, Spain, and Turkey.

This observational, cross-sectional quantitative study was conducted in accordance with the STROBE (Strengthening the Reporting of Observational Studies in Epidemiology) [43] guidelines. Participation was voluntary and anonymous. Respondents were eligible for inclusion if they met one of the predefined profiles confirmed within the questionnaire:Group P (dysphagia-related professionals): healthcare professionals (healthcare specialists—including medical doctors and clinical nutritionists—D-Ns, nursing specialists, and SLPs), non-health professionals involved in dysphagia care (support workers specialized in dysphagia, cooks, food technologists), and teachers or university students in their final years of degrees directly related to the above fields. For students and teachers, belonging to a relevant field was confirmed by requesting their academic discipline.Group non-P (dysphagia-related non-professionals): adults with dysphagia and/or swallowing disorders, family members and caregivers of adults or children with dysphagia, and other people interested in this disorder (for example, members or volunteers of organizations that specifically support people with dysphagia).

Exclusion criteria were applied to remove responses from participants who (a) did not match any of the established profiles, (b) provided incomplete data on the main study variables (self-perceived knowledge level and learning needs), or (c) represented profiles with insufficient sample sizes in their respective countries (*n* < 5).

### 2.2. Questionnaire Development and Validation

Two short questionnaires (one per target group) were developed in English by the project research team. The development process adhered to CHERRIES (Checklist for Reporting Results of Internet E-Surveys) [44] recommendations for online survey transparency and validation. The initial versions were reviewed by three experts in dysphagia management for face and content validity, evaluating item clarity, relevance, and comprehensiveness. Revisions were made accordingly to improve question wording and content coverage. A pilot test with 12 participants (6 per target group) assessed usability and comprehension, leading to minor adjustments in terminology and layout. Final questionnaires were translated into Greek, Italian, Turkish, Lithuanian, and Spanish using a forward–backward translation procedure following Eremenco et al. (2005) [45], with semantic and conceptual equivalence verified by bilingual experts.

Both questionnaires consisted of three sections: (1) Participant characteristics (age, sex, country, profile, profession or field of study, and personal or professional experience with dysphagia); (2) Self-perceived knowledge level and learning needs regarding dysphagia; and (3) Opinions on relevant dysphagia thematic areas and preferences for learning methodologies and course characteristics (format, duration, and preferred materials). The thematic areas were defined by consensus among project experts and grouped into four blocks: (a) “Identification & Treatment”—definition, detection and diagnosis, pharmacological and surgical treatment, active/clinical management procedures; (b) “Care & Feeding”—oral hygiene, medication administration, feeding techniques, equipment and mealtime environment; (c) “Diet & Nutrition”—malnutrition and dehydration prevention, nutritional adequacy, balanced diet; (d) “Food Preparation”—texture-modified foods, commercial thickeners, safe and appealing recipes. Question formats included closed-ended multiple-choice items (single-answer and check-all-that-apply), yes/no questions, and Likert scales.

### 2.3. Data Collection, Recoding of Variables and Statistical Analysis

A non-probabilistic snowball sampling approach was used to recruit participants across Greece, Italy, Lithuania, Spain, and Turkey. Each national team initially distributed the online survey through institutional and professional networks and encouraged recipients to forward it to relevant contacts. This process yielded 513 responses, corresponding to an overall estimated multiplication factor of 1.8 between invitations and responses (see Figure 1).

Questionnaires were administered via Google Forms^®^ (March–April 2021) in accordance with the procedure applied in previous publications [46]. To prevent duplicate submissions, Google Forms’ single-response restriction per user account was enabled.

For analysis, the 1–10 scales for self-perceived knowledge level and learning needs were recoded for easier interpretation and group comparison: 1–4 = low, 5–6 = medium, 7–10 = high. A 4/2/4 distribution was selected instead of equal intervals to more strictly differentiate between high and low levels, maintaining a narrow middle range as medium.

Descriptive statistics (frequencies and percentages) were calculated for all variables. Fisher’s exact tests were used to assess differences between two groups, and Chi-square tests were used to explore differences between several groups. For significant Chi-square tests, standardized residuals were examined to identify which cells showed statistically significant differences from expected frequencies. Statistical analyses were performed using SPSS v26.0 (IBM Corp., Armonk, NY, USA), and significance was set at *p* < 0.05.

### 2.4. Ethical and Data-Protection Considerations

Because this study involved anonymous online surveys without personal or sensitive data, ethical committee approval was not required in accordance with institutional and European regulations for minimal-risk research. However, questionnaires were validated by the Data Protection Unit of the University of Zaragoza to ensure compliance with the General Data Protection Regulation (GDPR) as implemented in Spain through Organic Law 3/2018 on Personal Data Protection and Digital Rights.

All participants were informed of the study purpose, data handling procedures, and their right to withdraw at any time before providing consent.

## 3. Results

### 3.1. Participants Characteristics

Initially, 304 responses were collected from Group P and 209 from Group non-P. After applying exclusion criteria, 7 cases from Group P and 43 from Group non-P were removed. The final sample comprised 297 participants from Group P and 166 from Group non-P (see Figure 1).

As it is shown in Table 1 Group P comprised health professionals (21% nurses, 15% D-Ns, 15% healthcare specialists, and 8% SLPs), non-health professionals (9% specialists working with patients with dysphagia in institutions, and 2% cook or food industry specialists), University/College lecturers (9%), and University/College students (20%). 40% of the responses came from Spain while the remaining responses (60%) were evenly distributed among Greece, Italy, Lithuania, and Turkey. The respondents were predominantly women (77%) while age was evenly distributed within the ranges. When they were asked about the sources they learned about this disorder, the two most frequently selected options were “At school/studies” (57%), meaning formal academic education, and “In the work environment” (35%), indicating practical workplace experience. Other sources, such as “Courses or training” and “On its own initiative,” yielded relatively lower response rates (24% in both cases).

Table 2 presents the description of group non-P. This group included adults with dysphagia (19%), family members (25%) or caregivers (30%) of a person with dysphagia, as well as other people interested in this condition (27%). A majority of this group were women (66%) with a homogeneous representation in terms of age, except for the older group (65–79 years old), which presented a lower number of responses (11%). Regarding the use of compensatory procedures for the management of dysphagia, 85% of the respondents (excluding other people interested in dysphagia) selected dysphagia-adapted dishes and beverages, mainly texture-modified foods (35%) and, to a lesser extent, thickened liquids (14%), using commercially available texture-modified foods and thickened beverages (15%), with 20% of them indicating several options. It is remarkable that only 19% of the respondents answered affirmatively to the question “Have you ever attended a training about dysphagia?”.

### 3.2. Self-Perceived Level of Knowledge and Learning Needs

As shown in Table 3, 59% of Group P considered that their knowledge about dysphagia was “Good/Excellent” whereas only 14% rated it as “Poor/Very poor. Statistically significant differences were found among profiles, with a higher percentage of “Good/Excellent” ratings among D-Ns (73%) and a lower one among students (47%), along with a lower percentage of nurses indicating the option “Poor/Very poor” (5%) compared to the highest values, which were observed in university/college students (30%) (*p* = 0.003). Table 4 reveals that Group non-P participants predominantly rated their dysphagia knowledge as limited, with 53% considering it “Poor/Very poor” compared to only 8% who rated it as “Good/Excellent”. No statistically significant differences were found among the different profiles within Group non-P (*p* = 0.189).

Statistical analysis confirmed that the observed differences between Group P and Group non-P in self-perceived knowledge levels were significant (*p* < 0.001).

Self-perceived education and training needs for dysphagia management among Group P are presented in Table 5. Most participants (62%) selected “Very probably/Definitely,” while only 12% indicated they did not require additional knowledge and skills in dysphagia. Intra-group analysis revealed statistically significant differences (*p* = 0.008), with D-Ns and university/college students showing the highest educational needs (80%). Table 6 shows the self-perceived education and training needs among Group non-P participants. Again, most respondents (69%) indicated “Very probably/Definitely” needing additional training, while only 12% felt they did not require further knowledge and skills in dysphagia management. In contrast to Group P, no statistically significant differences were found among the different profiles within Group non-P (*p* = 0.131).

However, when comparing the overall needs between both groups, no significant differences were found (*p* = 0.375), suggesting that professional and non-professional stakeholders share similar perceptions regarding their dysphagia learning requirements.

Finally, an exploratory analysis examined whether participants’ experience with dysphagia patients or personal caregiving was associated with their self-perceived knowledge levels and educational and training needs. As presented in Table 7, 70% of those reporting experience considered that their knowledge on dysphagia was “Good/Excellent” compared to 49% of those reporting no experience (*p* = 0.013). However, no statistically significant differences were found between both groups regarding self-perceived need for education and training, with 61% and 57%, respectively, selecting “Very probably/Definitely” (*p* = 0.216).

### 3.3. Relevant Dysphagia Thematic Areas and Preferences for Learning Methodologies

Figure 2 presents the perceived relevance of different dysphagia thematic areas among all participants. “Identification & Treatment” was considered relevant by 63% of respondents, while the remaining three areas (“Care & Feeding,” “Diet & Nutrition,” and “Food Preparation”) were selected by approximately 50% each. Only in the area “Identification & Treatment” did people from Group non-P find that they needed more knowledge (72%) than professionals (58%) (*p* = 0.002). Interest was similar for the remaining areas, with response percentages between 47% and 53%.

Table 8 and Table 9 display the perceived relevance of dysphagia thematic areas for Group P and Group non-P, respectively. Only Group P displayed statistically significant intra-group differences in interest in the different areas (*p* < 0.001 in all cases). “Identification & Treatment” was chosen more frequently by SLPs (92%) and healthcare specialists (72%) than by D-Ns (30%). On the other hand, D-Ns were those who expressed the greatest interest in “Care & Feeding” (68%), compared to healthcare specialists (28%) and university lecturers (29%), who expressed less interest. D-Ns were those who expressed a greater degree of preference for the topic “Diet & Nutrition” (86%), whereas healthcare specialists and SLPs selected that option less frequently (35% and 20%, respectively). Once again, D-Ns were the ones with the highest interest in “Food Preparation” (93%) as compared to other health professionals, such as nursing specialists (41%) and healthcare specialists (21%).

Concerning what would be regarded as an acceptable duration of a possible learning program, most participants expressed a preference for short-term training learning courses: 76% favored the option of a course lasting less than 20 h (Figure 3).

This was especially evident among the members of Group non-P, in which significant differences could be observed compared to Group P in an expressed preference for training lasting less than 10 h (*p* = 0.030).

As depicted in Figure 4, the preferred answers to the question “What learning resources would be most effective for you?” were the following: short explanatory videos (75%), practical exercises (55%), the use of images, graphics and illustrations (48%), PowerPoint presentations (37%).

Less popular choices, selected by fewer than 25% of participants, included evaluation tests and written exercises. A comparison of responses between the two surveyed groups revealed statistically significant differences only for practical exercises (*p* = 0.003) and the use of tests (*p* = 0.013). In both instances, a higher percentage of respondents in Group non-P opted for each of those respective options.

## 4. Discussion

This observational, cross-sectional quantitative study aimed to provide a comprehensive overview of the current self-perceived knowledge and learning needs among two multidisciplinary groups—professionals (Group P) and non-professionals (Group non-P)—based on participants’ self-perceptions rather than objective performance measures. The inclusion of healthcare and non-healthcare professionals, as well as patients, relatives, and caregivers, reflects the complexity of dysphagia management. Both groups were predominantly female, consistent with the gender distribution reported in prior studies on health professionals and caregivers [47].

In Group non-P, most participants rated their knowledge as poor or very poor, and 81% reported no prior training. This result aligns with previous research indicating that insufficient caregiver knowledge can hinder adherence to dietary recommendations [48,49,50] and increase perceived burden [51]. Education is valued by caregivers, who often report that improved understanding of dysphagia reduces stress and improves their ability to provide effective support [51,52]. While studies directly exploring patients’ self-perceived knowledge are scarce, findings in individuals with Parkinson’s disease or neuromuscular disorders show that patients and their families often wish to learn more about dysphagia and its treatment [36,53].

In contrast, most professionals in Group P rated their knowledge as good or excellent, although this varied significantly by profession, with D-Ns reporting the highest scores. These differences likely reflect disparities in access to training, professional responsibilities, and country-specific practices [54]. Notably, despite these higher ratings, many professionals—especially D-Ns and nurses—expressed a strong desire for further education. This apparent contradiction suggests that self-reported “good” knowledge may reflect confidence in basic concepts, but not necessarily readiness for complex, real-world clinical scenarios. Similar patterns have been reported in other settings, where professionals acknowledge training gaps despite high self-assessments [4,16,18,20,23,55]. Given the exploratory design of the present study, it was not possible to identify predictors of these differences through multivariate analysis; however, this represents a relevant direction for future research. Recent international evidence further supports this interpretation: the TEAMS survey found that healthcare professionals across multiple countries frequently overestimate their competence in malnutrition and dysphagia management, while still identifying substantial needs for targeted education [25].

Educational needs in Group P were closely linked to professional roles: SLPs and healthcare specialists prioritized “Identification & Treatment,” whereas D-Ns showed greater interest in diet, nutrition, and food adaptation topics. Nurses expressed interest in all areas, reflecting their broad responsibilities in dysphagia care [16,17,24]. In Group non-P, “Identification & Treatment” also emerged as the top priority, followed by diet-related and food preparation topics. This likely reflects a desire to better recognize symptoms, understand the condition, and improve day-to-day management. These findings are consistent with reviews noting insufficient nutritional knowledge among caregivers and the general public [56].

A key finding is that both groups, regardless of self-perceived knowledge level, expressed a clear need for further education and training. This indicates that the decision to seek training is influenced not only by awareness of knowledge gaps, but also by recognition of the complexity of dysphagia management and the need for up-to-date skills. Previous studies support the importance of continuous professional development and tailored education for both healthcare and non-healthcare participants in dysphagia care [16,20,23,57]. The role of caregivers, family members, and acquaintances in early recognition and daily management of dysphagia is crucial; however, this collective often faces significant challenges due to limited knowledge and insufficient training [31,32].

The literature on dysphagia education for non-healthcare professionals—such as chefs, catering managers, and food technologists—remains limited; however, these roles are essential for implementing safe dietary modifications. Studies in the United States have found significant knowledge gaps in food service workers and trainees regarding dysphagia and texture-modified diets [27,58]. Recent advances in dysphagia research, including rheological analysis of modified foods and the implementation of standardized texture levels [26,59,60,61], reinforce the need for targeted education that spans clinical and food science disciplines. Interventions involving food service staff, nurses, and healthcare assistants have shown positive impacts on knowledge and compliance with texture-modified diets [62], but such opportunities are far less available for non-healthcare workers [20,29].

Regarding training formats, our results are consistent with evidence that suggests engagement, interactivity, and concise content are crucial for uptake [28,63]. Consequently, these learning needs could be addressed through technology-enhanced educational interventions. Digital health technologies, including e-learning platforms, mobile health applications, and simulation-based training tools, offer accessible, flexible, and scalable solutions for delivering dysphagia education to diverse stakeholder groups [64]. For healthcare professionals, online continuing education modules and virtual simulation tools provide practice-based learning without time and location constraints [65]. For patients and caregivers, mobile applications and web-based resources could offer practical guidance on safe feeding strategies, texture-modified diets, and symptom recognition in accessible formats [66]. In summary, well-designed digital tools incorporating interactive technologies—such as artificial intelligence-driven personalization, telemonitoring, or Internet of Things-based feedback systems—could further enhance educational delivery and patient management [64]. However, the design and implementation of such technology-based interventions must be grounded in adult learning principles [67] and rigorously evaluated to ensure effectiveness across different user groups and contexts [68].

### Limitations and Future Research

This study’s strengths include the variety of profiles of participants obtained thanks to the recruitment methodology, which allowed for comparisons between professional and non-professional groups and across different professional roles. However, the use of a convenience and snowball sampling strategy resulted in unequal representation across participating countries, with a predominance of respondents from Spain. Although this method does not allow for calculating an exact response rate, it was considered appropriate to reach a wide profile of stakeholders. Differences between professional groups observed in this study may reflect variations in access to training, professional responsibilities, and contextual factors among countries. Nevertheless, due to sample size and exploratory design, country-level comparisons and multivariate analyses to identify predictors were not feasible. Future studies with larger and more homogeneous samples are warranted to identify potential predictors of dysphagia-related knowledge and training needs, as well as to explore international variability in educational priorities.

Self-reported measures, which are widely used in exploratory studies [40], may be influenced by recall or social desirability bias, and perceived knowledge may not accurately reflect actual competence. Future research should use validated knowledge assessment tools (e.g., KAP or KAB frameworks) and objective quizzes in combination with consensus methods such as Delphi studies to identify priority topics, optimal learning modalities, and appropriate competence assessment tools for each group.

## 5. Conclusions

This international observational cross-sectional study reveals that both professionals and non-professionals involved in dysphagia management express a strong need for further learning, despite marked differences in self-perceived knowledge levels. Among professionals, self-reported “good” knowledge often coexists with recognition of the need for advanced and updated skills, whereas among non-professionals, limited baseline knowledge drives the desire for fundamental understanding of the condition and how to manage day-to-day care. Educational and training priorities vary significantly across professional roles: speech and language pathologists and healthcare specialists emphasize “Identification & Treatment,” dietitians and nutritionists focus on “Diet & Nutrition” and “Food Preparation,” while nurses show interest across all areas. In the non-professional group, interest was consistently high across themes, particularly for “Identification & Treatment”, highlighting the need for foundational competence and confidence in recognizing and managing dysphagia. These findings underscore the importance of developing tailored and inclusive learning programs that address role-specific competencies while incorporating accessible and engaging formats. Short, visually rich, and interactive e-learning modules may be particularly effective for enhancing engagement and retention. Incorporating emerging digital and assistive technologies could further enhance these programs, ensuring effective dissemination, sustained engagement, and translation into improved dysphagia care and quality of life. Additionally, as discussed, future studies with larger and more balanced samples across countries are warranted to identify predictors of dysphagia knowledge and to support cross-cultural comparisons, thereby refining educational strategies at the European level.

## Figures and Tables

**Figure 1 healthcare-13-03140-f001:**
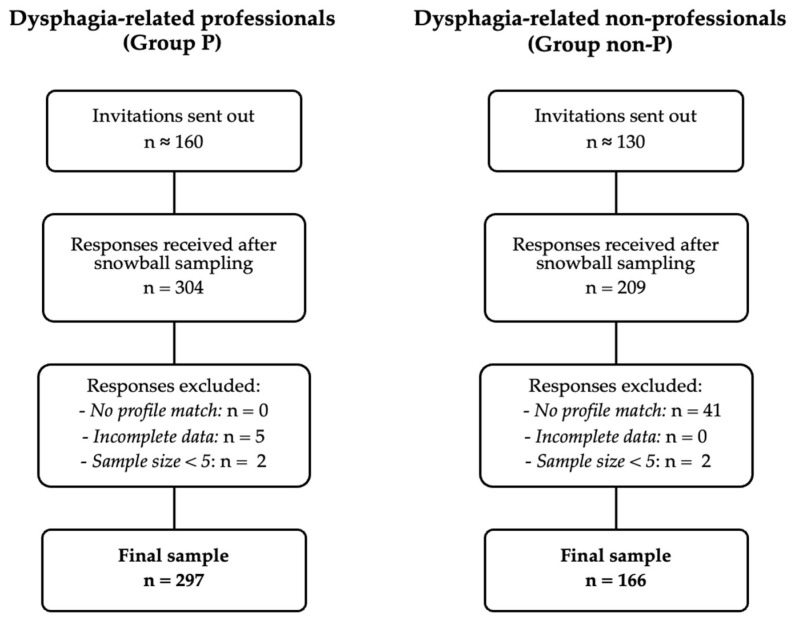
Flow survey participants.

**Figure 2 healthcare-13-03140-f002:**
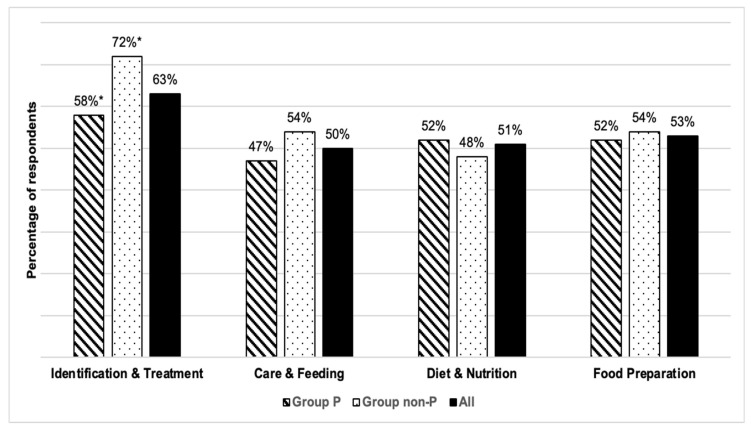
Relevant dysphagia thematic areas according to survey participants. Note: * Statistically different results, *p*-value = 0.002, by Fisher’s exact test.

**Figure 3 healthcare-13-03140-f003:**
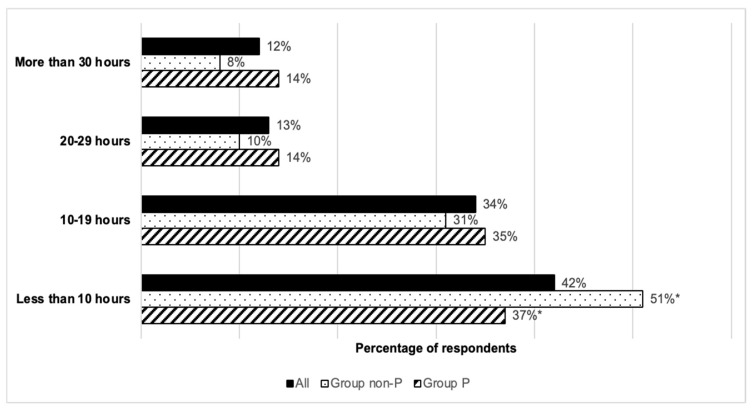
Preferences regarding the length of the learning courses. Note: * Statistically different results, *p*-value = 0.030, by Fisher’s exact test.

**Figure 4 healthcare-13-03140-f004:**
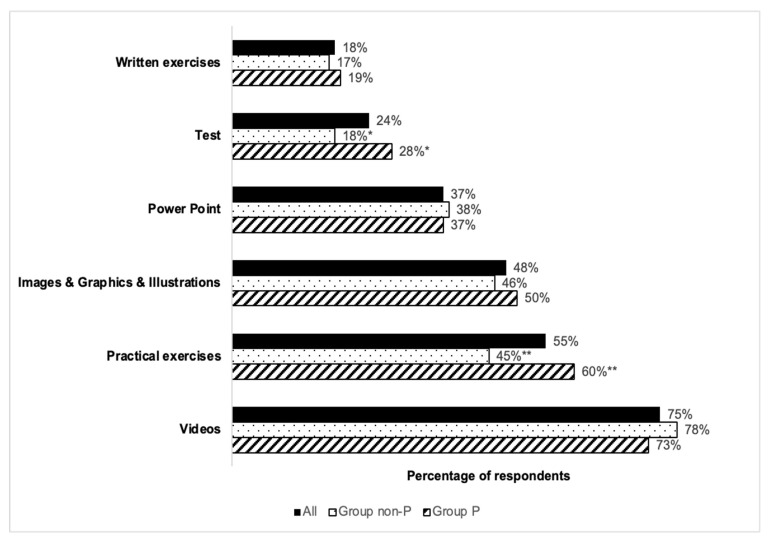
Preferences regarding the educational resources to be used in learning courses. Note: Statistically different results, * *p*-value = 0.013 and ** *p*-value = 0.003, by Fisher’s exact test.

**Table 1 healthcare-13-03140-t001:** Description of dysphagia-related professionals’ group (Group P) (*n* = 297).

Country	Greece	38 (13)
	Italy	42 (14)
	Lithuania	45 (15)
	Spain	120 (40)
	Turkey	52 (18)
Gender	Male	65 (22)
	Female	230 (77)
	Rather not say	2 (1)
Age	18–29 years old	114 (38)
	30–45 years old	113 (38)
	46–64 years old	70 (24)
Profile	Healthcare professionals	175 (59)
	Dietitian-Nutritionists	44 (15)
	Healthcare specialists	43 (15)
	Nursing specialists	63 (21)
	Speech-language pathologists	25 (8)
	Non-healthcare professionals	34 (11)
	Specialist in an institution	27 (9)
	Cook/Food industry specialists	7 (2)
	University/College lecturers	28 (9)
	University/College students	60 (20)
What sources did you learn about dysphagia from? ^a^	At school/studies	168 (57)
	In the work environment	105 (35)
	In courses/training	71 (24)
	Own initiative	70 (24)
	Other sources	0 (0)

Data expressed as *n* (%). ^a^ More than one option could be chosen.

**Table 2 healthcare-13-03140-t002:** Description of dysphagia-related non-professionals’ group (Group non-P) (*n* = 166).

Country	Greece	18 (11)
	Italy	22 (13)
	Lithuania	27 (16)
	Spain	55 (33)
	Turkey	44 (27)
Gender	Male	56 (34)
	Female	106 (64)
	Rather not say	4 (2)
Age	<18–29 years old	49 (30)
	30–45 years old	60 (36)
	46–64 years old	38 (23)
	65–79 years old	19 (11)
Profile	Adult with dysphagia	32 (19)
	Family member of a person with dysphagia	41 (25)
	Family member of a child	21 (13)
	Family member of an adult	20 (12)
	Caregiver of a person with dysphagia	49 (30)
	Caregiver of a child	11 (7)
	Caregiver of an adult	38 (23)
	Other people interested in dysphagia	44 (27)
Do you/Does the person you care for/use …? ^a,b^	Texture-modified foods	41 (35)
	Thickened liquids	17 (14)
	Commercial products	18 (15)
	A combination of the above	24 (20)
	None of the above	18 (15)
Have you ever attended training about dysphagia?	Yes No	31 (19) 134 (81)

Data expressed as *n* (%). ^a^ More than one option could be chosen. ^b^ “Other people interested in dysphagia” did not answer this question.

**Table 3 healthcare-13-03140-t003:** Self-perceived level of knowledge of dysphagia-related professionals (Group P) (*n* = 297).

	Very Poor/Poor	Fair	Good/Excellent
Dietitian-Nutritionists	3 (7)	9 (21)	32 (73) *
Healthcare specialists	4 (9)	11 (26)	28 (65)
Nursing specialists	3 (5) *	25 (40) *	35 (56)
Speech-language pathologists	5 (20)	7 (28)	13 (52)
Non-healthcare professionals	6 (18)	6 (18)	22 (65)
University/College lecturers	2 (7)	10 (36)	16 (57)
University/College students	18 (30) *	14 (23)	28 (47) *
All	41 (14)	82 (28)	174 (59)

Notes: Values are expressed as number (percentage). *p*-value = 0.003, by Pearson chi-square test. * Indicates cells showing statistically significant differences from expected frequencies.

**Table 4 healthcare-13-03140-t004:** Self-perceived level of knowledge of dysphagia-related non-professionals (Group non-P) (*n* = 153).

	Very Poor/Poor	Fair	Good/Excellent
Adult with dysphagia	20 (67)	9 (30)	1 (3)
Family member of a person with dysphagia	20 (50)	19 (48)	1 (3)
Caregiver of a person with dysphagia	21 (48)	16 (36)	7 (16)
Other people interested in dysphagia	20 (51)	16 (41)	3 (8)
All	81 (53)	60 (39)	12 (8)

Notes: Values are expressed as number (percentage). *p*-value = 0.189, by Pearson chi-square test. Group non-P’s participants who answered “Don’t know” are not included.

**Table 5 healthcare-13-03140-t005:** Self-perceived education and training needs of dysphagia-related professionals (Group P) (*n* = 297).

	No Need	Possibly/Probably	Very Probably/Definitely
Dietitian-Nutritionists	4 (9)	5 (11) *	35 (80) *
Healthcare specialists	8 (19)	12 (28)	23 (54)
Nursing specialists	7 (11)	22 (35)	34 (54)
Speech-language pathologists	3 (12)	10 (40)	12 (48)
Non-healthcare professionals	5 (15)	8 (24)	21 (62)
University/College lecturers	6 (21)	10 (36)	12 (43) *
University/College students	4 (7)	8 (13) *	48 (80) *
All	37 (12)	75 (25)	185 (62)

Notes: Values are expressed as number (percentage). *p*-value = 0.008, by Pearson chi-square test. * Indicates cells showing statistically significant differences from expected frequencies.

**Table 6 healthcare-13-03140-t006:** Self-perceived education and training needs of dysphagia-related non-professionals (Group non-P) (*n* = 166).

	No Need	Possibly/Probably	Very Probably/Definitely
Adult with dysphagia	5 (16)	7 (23)	19 (61)
Family member of a person with dysphagia	1 (2)	10 (24)	30 (73)
Caregiver of a person with dysphagia	4 (8)	7 (14)	38 (78)
Other people interested in dysphagia	9 (21)	9 (21)	26 (59)
All	19 (12)	33 (20)	113 (69)

Notes: Values are expressed as number (percentage). *p*-value = 0.131, by Pearson chi-square test. Group non-P’s participants who answered “Don’t know” are not included.

**Table 7 healthcare-13-03140-t007:** Self-perceived level of knowledge and education and training needs compared across experienced and no experienced dysphagia-related professionals ^a^ (*n* = 209).

		Experience		
		Yes (*n* = 132)	No (*n* = 77)	*p*
Gender	Male	24 (18)	17 (22)	0.305 ^b^
	Female	108 (82)	60 (78)	
Age	18–29 years	23 (17) *	30 (39) *	0.001 ^c^
	30–45 years	73 (55) *	26 (34) *	
	46–64 years	36 (27)	21 (27)	
Profile	D-N	27 (21)	17 (22)	0.085 ^c^
	Healthcare specialist	21 (16)	22 (29)	
	Nursing specialist	44 (33)	19 (25)	
	SLP	14 (11)	11 (14)	
	Non-health professionals	26 (20)	8 (10)	
Self-perceived level of knowledge about dysphagia	Very poor/Poor	10 (8)	11 (14)	0.013 ^c^
	Fair	30 (23) *	28 (36) *	
	Good/Excellent	92 (70) *	38 (49) *	
Self-perceived need for education and training on dysphagia	No need	13 (10)	14 (18)	0.216 ^c^
	Possibly/Probably	38 (29)	19 (25)	
	Very probably/Definitely	81 (61)	44 (57)	

Notes: Values are expressed as number (percentage) ^a^ University/College lecturers and students not considered. Comparisons by ^b^ Fisher’s exact test or ^c^ Chi^2^ Pearson. * Indicates cells showing statistically significant differences from expected frequencies.

**Table 8 healthcare-13-03140-t008:** Relevant dysphagia thematic areas for dysphagia-related professionals (Group P) (*n* = 297).

	Identification & Treatment	Care & Feeding	Diet & Nutrition	Food Preparation
Dietitian-Nutritionists	13 (30) *	30 (68) *	38 (86) *	41 (93) *
Healthcare specialists	31 (72) *	12 (28) *	15 (35) *	9 (21) *
Nursing specialists	36 (57)	36 (57)	32 (51)	26 (41) *
Speech-language pathologists	23 (92) *	8 (32)	5 (20) *	10 (40)
Non-healthcare professionals	20 (59)	17 (50)	17 (50)	17 (50)
University/College lecturers	13 (46)	8 (29) *	13 (46)	12 (43)
University/College students	35 (58)	29 (48)	35 (58)	40 (67) *

Notes: Values are expressed as number (percentage). *p*-value < 0.001 in all thematic areas, by Pearson chi-square test. * Indicates cells showing statistically significant differences from expected frequencies.

**Table 9 healthcare-13-03140-t009:** Relevant dysphagia thematic areas for dysphagia-related non-professionals (Group non-P) (*n* = 166).

	Identification & Treatment	Care & Feeding	Diet & Nutrition	Food Preparation
Adult with dysphagia	26 (81)	17 (53)	15 (47)	16 (50)
Family member of a person with dysphagia	26 (63)	25 (61)	18 (44)	20 (49)
Caregiver of a person with dysphagia	34 (69)	29 (59)	26 (53)	27 (55)
Citizen interested in dysphagia	34 (77)	18 (41)	20 (46)	26 (59)

Notes: Values are expressed as number (percentage).

## Data Availability

The data presented in this study are available on request from the corresponding author. All data relevant to the analysis are already included within the manuscript, while the raw files are not publicly distributed due to format limitations.

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
