# Peer review of "Knowledge Levels and Learning Needs in Dysphagia Management: Perspectives from Professional and Non-Professional Stakeholders in Five European Countries"

_healthcare, 2025, doi:10.3390/healthcare13233140_

Round 1

Reviewer 1 Report

Comments and Suggestions for Authors

Thank you for the opportunity to review this paper.

In the abstract, percentage values should not be included in the introduction without bibliographical references. It is suggested that percentage values be removed. The abstract is to long. The type of study performed should be indicated in the methodology.

The title must be aligned with the study objectives. The title is " Addressing Learning Needs in Dysphagia Management: Perspectives from Professional and Non-Professional Stakeholders", when we read the title we think that the study is focused in learning needs, but the aim is focused in assess self-perceived knowledge levels and learning. This is quite different. I suggest that the title could be: Knowledge levels and learning needs in Dysphagia Management: Perspectives from Professional and Non-Professional Stakeholders The aim described at the end of the introduction must be the same as that identified in the abstract.

The study has methodological weaknesses, as the study type is not indicated. The study should follow the EQUATOR guidelines.

The mention of the use of ChatGPT (GPT-5) is transparent and positive, demonstrating a commendable attitude of scientific integrity by the authors. However, to ensure compliance with the editorial best practices recommended by COPE and ICMJE, it is important that the role of the model be clearly described, specifying that no unverified scientific content was generated, that the tool was used solely as a writing and stylistic aid, and that the final control and full review of the content were carried out by the human authors

It is important indicated the alpha the Cronbach, to more statistic fiability.

It is relevant to do multivariate statistic analysis to identify possibles preditors of Knowledge.

The sample was obtained through convenience and snowball sampling, without information on the total number of invitations sent or the participation rate. It is therefore recommended to include an estimate of the potential number of participants contacted.

Author Response

Response to Reviewer 1 Comments:

Thank you very much for taking the time to review this manuscript. Please find the detailed responses below and the corresponding revisions/corrections highlighted in red in the re-submitted files

Point-by-point response to Comments and Suggestions for Authors:

  • Comment 1: In the abstract, percentage values should not be included in the introduction without bibliographical references. It is suggested that percentage values be removed. The abstract is to long. The type of study performed should be indicated in the methodology.

Response 1: Thank you for pointing this out. We agree with this comment. We have, accordingly, revised the whole abstract to correct this point following your recommendations

  • Comment 2: The title must be aligned with the study objectives. The title is " Addressing Learning Needs in Dysphagia Management: Perspectives from Professional and Non-Professional Stakeholders", when we read the title we think that the study is focused in learning needs, but the aim is focused in assess self-perceived knowledge levels and learning. This is quite different. I suggest that the title could be: Knowledge levels and learning needs in Dysphagia Management: Perspectives from Professional and Non-Professional Stakeholders.

Response 2: Thank you for pointing this out. We agree with this comment. We have, accordingly, modified the title.

  • Comment 3: The aim described at the end of the introduction must be the same as that identified in the abstract.

Response 3: Thank you for your suggestion. We have revised the wording of the study aim in the abstract to ensure it aligns precisely with the aim stated in the introduction

  • Comment 4: The study has methodological weaknesses, as the study type is not indicated. The study should follow the EQUATOR guidelines.

Response 4: We appreciate the reviewer’s observation regarding the need to specify the study type and adherence to reporting standards. The study type has now been clearly indicated in the Abstract and Materials and Methods section as an observational, cross-sectional quantitative study. In addition, we have clarified in the Materials and Methods section that this study follows the STROBE (Strengthening the Reporting of Observational Studies in Epidemiology) and CHERRIES (Checklist for Reporting Results of Internet E-Surveys) guidelines. To ensure full transparency and methodological rigor, we have also included the corresponding checklists for both frameworks as supplementary material.

  • Comment 5: The mention of the use of ChatGPT (GPT-5) is transparent and positive, demonstrating a commendable attitude of scientific integrity by the authors. However, to ensure compliance with the editorial best practices recommended by COPE and ICMJE, it is important that the role of the model be clearly described, specifying that no unverified scientific content was generated, that the tool was used solely as a writing and stylistic aid, and that the final control and full review of the content were carried out by the human authors.

Response 5: Thank you for this valuable observation regarding the transparency and appropriate disclosure of AI tool usage. We completely agree with the importance of adhering to COPE and ICMJE best practices. In response to your recommendation, we have revised the Acknowledgments section to clearly specify that ChatGPT (GPT-5) was used solely as a writing and stylistic aid, that no unverified scientific content was generated, and that the authors maintain full responsibility for critically reviewing, editing, and validating all content. The revised statement now explicitly confirms that final control and complete review of the manuscript content were carried out by the human authors, ensuring full compliance with editorial guidelines and maintaining the scientific integrity of our work.

  • Comment 6: It is important indicated the alpha the Cronbach, to more statistic fiability.

Response 6: We appreciate the reviewer's suggestion regarding statistical reliability. However, Cronbach's alpha is not appropriate for our questionnaire because we do not intend to measure a unitary construct or calculate a total score. Each question evaluates distinct and independent aspects of self-perceived knowledge levels and learning needs, and they were analyzed separately according to their nature. As recent methodological studies indicate (Edelsbrunner, 2024; Hussey et al., 2025), Cronbach's alpha is only relevant when items are reflective indicators of the same latent construct. Our approach aligns with a formative) model where items do not need to be correlated with each other (Aguirre-Urreta & Rönkkö, 2024)

  • Aguirre-Urreta MI & Rönkkö M. Reconsidering the implications of formative versus reflective measurement model misspecification. Inf Syst J. 2024;34:533–584. https://doi.org/10.1111/isj.12487
  • Edelsbrunner, P.A., Simonsmeier, B.A. & Schneider, M. The Cronbach’s Alpha of Domain-Specific Knowledge Tests Before and After Learning: A Meta-Analysis of Published Studies. Educ Psychol Rev 37, 4 (2025). https://doi.org/10.1007/s10648-024-09982-y
  • Hussey I, Alsalti T, Bosco F, Elson M, Arslan R. An Aberrant Abundance of Cronbach’s Alpha Values at .70. Adv Methods Pract Psychol Sci. 2025;8(1). https://doi.org/10.1177/25152459241287123
  • Comment 7: It is relevant to do multivariate statistic analysis to identify possibles preditors of Knowledge.

Response 7: We thank the reviewer for this suggestion. However, this study was designed as an exploratory investigation with the primary aim to provide a comprehensive overview of the current self-perceived knowledge and learning needs among stakeholders involved in dysphagia management, rather than as an explanatory study to identify predictors. Conducting multivariate predictive analyses would require a different theoretical framework, specific hypotheses, and a study design optimized for that purpose. We believe that identifying predictors of knowledge and/or learning needs represents an important direction for future research, which we have now addressed in the discussion as part of the “Limitations and Future Research” subsection

  • Comment 8: The sample was obtained through convenience and snowball sampling, without information on the total number of invitations sent or the participation rate. It is therefore recommended to include an estimate of the potential number of participants contacted.

Response 8: We thank the reviewer for this suggestion. We have now included a flow diagram (Figure 1) in the Results section that illustrates the participant recruitment and selection process, including the number of participants contacted, number of responses received, exclusions applied, and the final sample analyzed. Material and Methods section has been updated to indicate the number of survey invitations sent and the corresponding multiplication factor obtained through the snowball sampling procedure. This visual and textual information enhances the transparency and clarity of our sampling procedure.

Reviewer 2 Report

Comments and Suggestions for Authors

Dear Authors, the study required updates for possible international consideration. The tile required clarity: I suggest to consideder to insert the type of stuyd conducted and setting. I suggest streamlining the Abstract to the conventional format of no more than 250 words and making the Conclusions more impactful in terms of clinical and assitive practice (see the disucussion too). Currently, they remain very general and overly focused on the results obtained. Keywords: see the comments for the title for setting and type of study conducted and always rest in max. 4/5 words. I recommend being clearer regarding the study objectives (lines 91-98), which are currently not easily understandable. Methods: the part that certly required major attention. I suggest to adopt international reporting tool (with the references and the check list in the supplementary files) mandatary for scientific community and could help the authors to present the manuscript in international and scientific perspective (e.g. Equator Network: https://www.google.com/search?client=firefox-b-d&q=equator+network). The correction of this elements, is funtamental for possible international consideratin. Missing relevant element that required clarity: recruitng, inclusion and exlusion criteria, and pre-test for validity of survey required major scientific details. The Results overall well presented but I suggest a possible Flow of the study. In the Discussion section, a real interpretation in terms of clinical and assistive practice is completely lacking, especially from new techonolgy view. In this regard, I suggest the authors consider adding a new section titled “Perspectives for Clinical and Assistive Practice”, in which they can discuss the role of new technologies for patient management across all phases of care, mainly for population considered. To strengthen this section, the authors could starting the discussion with appropriate references on topics such as The Internet of Things in the Nutritional Management of Patients with Chronic Neurological Cognitive Impairment, Applications of artificial intelligence in drug discovery for neurological diseases, and Artificial Intelligence in The Management of Neurodegenerative Disorders. These additions would significantly enrich the section and support a wider dissemination of the findings to a broader audience of readers and researchers. A dedicated section on limitations is needed and extend of generalizabilty of data finding, and the Conclusions will also require updates in accordance with the above-mentioned revisions. References: see the comments above and update the reference over 10 years (many of them are very old) if not for method section or relevant evidence based data finding. In summary, I suggest responding point by point to each individual suggestion for possible reconsideration, principally in clinical and assistive practice view and for the methods.

Comments on the Quality of English Language

Native review request

Author Response

Response to Reviewer 2 Comments:

Thank you very much for taking the time to review this manuscript. Please find the detailed responses below and the corresponding revisions/corrections in track changes in the re-submitted files

Point-by-point response to Comments and Suggestions for Authors:

  • Comment 1: The tile required clarity: I suggest to consideder to insert the type of stuyd conducted and setting.

Response 1: Thank you for your suggestion regarding the manuscript title. We appreciate your feedback and have revised the title to improve clarity and better align it with the manuscript content. Regarding the inclusion of the study type in the title, we respectfully believe this may be less necessary given that the study design is now clearly stated at the beginning of the abstract, where readers can immediately identify it. We believe this approach maintains clarity while avoiding an overly lengthy title.

  • Comment 2: I suggest streamlining the Abstract to the conventional format of no more than 250 words and making the Conclusions more impactful in terms of clinical and assitive practice (see the disucussion too). Currently, they remain very general and overly focused on the results obtained. Keywords: see the comments for the title for setting and type of study conducted and always rest in max. 4/5 words.

Response 2: Thank you for pointing this out. We agree with this comment. We have, accordingly, revised the whole abstract to correct this point.

  • Comment 3: I recommend being clearer regarding the study objectives (lines 91-98), which are currently not easily understandable.

Response 3: Thank you for this important observation. You are absolutely right that the study objectives lacked clarity in the original version. We have revised this paragraph to better reflect the exploratory nature of our study. Specifically, we replaced "assess" with "explore" to more accurately convey that we are examining self-perceived knowledge and opinions rather than conducting objective measurements. We have also clarified the final sentence to explicitly state how the findings could be used. We believe these changes significantly improve the transparency and understandability of our study aims.

  • Comment 4: Methods: the part that certly required major attention. I suggest to adopt international reporting tool (with the references and the check list in the supplementary files) mandatary for scientific community and could help the authors to present the manuscript in international and scientific perspective (e.g. Equator Network: https://www.google.com/search?client=firefox-b-d&q=equator+network). The correction of this elements, is funtamental for possible international consideratin. Missing relevant element that required clarity: recruitng, inclusion and exlusion criteria, and pre-test for validity of survey required major scientific details.

Response 4: We appreciate the reviewer’s detailed suggestions regarding the need to adopt international reporting tools. In response, we have thoroughly revised the Materials and Methods section, incorporating clarifications and additional details on study design, participant recruitment, inclusion/exclusion criteria, questionnaire pre-testing (face and content validity), and data collection procedures. We explicitly stated that this study follows both the STROBE (Strengthening the Reporting of Observational Studies in Epidemiology) and CHERRIES (Checklist for Reporting Results of Internet E-Surveys) guidelines. To ensure full transparency and methodological rigor, we also include the corresponding checklists for both frameworks as supplementary material. These revisions address the reviewer’s concerns and align the manuscript with international reporting standards for observational and survey-based research.

Regarding the questionnaire validation process, our approach (see section 2.3) was designed to be appropriate for an exploratory needs assessment study using ad-hoc questionnaires, rather than a psychometric scale validation study. This validation approach is consistent with recommended practices for developing questionnaires for exploratory needs assessment studies, where content validity and practical usability are the primary concerns. While our validation process followed established principles for needs assessment questionnaire development (expert review and pilot testing), we acknowledge that we did not document all procedural details with the level of specificity that would be ideal for reporting. We have now expanded the Methods section to provide all available detail regarding the validation process.

  • Comment 5: The Results overall well presented but I suggest a possible Flow of the study.

Response 5: We thank the reviewer for this suggestion. We have now included a flow diagram (Figure 1) in the Results section that illustrates the participant recruitment and selection process, including the number of responses received, exclusions applied, and the final sample analyzed. This visual representation enhances the transparency and clarity of our sampling procedure.

  • Comment 6: In the Discussion section, a real interpretation in terms of clinical and assistive practice is completely lacking, especially from new techonolgy view. In this regard, I suggest the authors consider adding a new section titled “Perspectives for Clinical and Assistive Practice”, in which they can discuss the role of new technologies for patient management across all phases of care, mainly for population considered. To strengthen this section, the authors could starting the discussion with appropriate references on topics such as The Internet of Things in the Nutritional Management of Patients with Chronic Neurological Cognitive Impairment, Applications of artificial intelligence in drug discovery for neurological diseases, and Artificial Intelligence in The Management of Neurodegenerative Disorders. These additions would significantly enrich the section and support a wider dissemination of the findings to a broader audience of readers and researchers.

Response 6: We appreciate the reviewer's suggestion to consider technological perspectives for addressing education and training needs. We have added a brief discussion in the Discussion section acknowledging that future research could explore how digital health technologies—such as online educational platforms, mobile applications, and simulation tools—might offer scalable solutions for dysphagia learning across different stakeholder groups. However, as our exploratory study focused specifically on assessing current self-perceived knowledge and learning needs rather than evaluating technological interventions, we have kept this addition concise to maintain the manuscript's focus while acknowledging promising future directions. Consequently, while the three references recommended by the reviewer (Sguanci et al. The Internet of Things in the Nutritional Management of Patients with Chronic Neurological Cognitive Impairment: A Scoping Review. Healthcare (Basel). 2024 Dec 25;13(1):23. doi: 10.3390/healthcare13010023; Ekins & Lane. Applications of artificial intelligence in drug discovery for neurological diseases. Neurotherapeutics. 2025 Jul;22(4):e00624. doi: 10.1016/j.neurot.2025.e00624; Dhankhar et al. Artificial Intelligence in The Management of Neurodegenerative Disorders. CNS Neurol Disord Drug Targets. 2024;23(8):931-940. doi: 10.2174/0118715273266095231009092603) are of high quality and considerable interest, they have not been incorporated into the manuscript as they address technological interventions beyond the scope of our current study.

  • Comment 7: A dedicated section on limitations is needed and extend of generalizabilty of data finding.

Response 7: We thank the reviewer for this important suggestion. Previously, limitations were briefly mentioned in the Discussion; we have now consolidated and expanded these into a dedicated Limitations and Future Research section at the end of the Discussion. We believe this addition strengthens the manuscript by providing readers with a clearer understanding of the study's scope and boundaries.

  • Comment 8: The Conclusions will also require updates in accordance with the above-mentioned revisions.

Response 8: We thank the reviewer for this observation. Conclusions section has been thoroughly revised to reflect all the modifications made throughout the manuscript in response to reviewers’ comments. The updated Conclusions section now clearly summarize the main findings, highlight differences in self-perceived knowledge between professional and non-professional stakeholders, emphasize role-specific educational priorities, and incorporate implications for the development of tailored learning programs. Additionally, the revised section acknowledges the potential of emerging digital and assistive technologies to enhance education and improve dysphagia care, ensuring that the conclusions are fully aligned with the updated Abstract, Discussion, and overall narrative of the manuscript.

  • Comment 9: References: see the comments above and update the reference over 10 years (many of them are very old) if not for method section or relevant evidence based data finding.

Response 9: We thank the reviewer for this observation. References have been revised to align with all the modifications made throughout the manuscript in response to the all reviewers' comments.

Reviewer 3 Report

Comments and Suggestions for Authors

This is a well-written, structured, and relevant manuscript addressing a clear and underexplored topic: cross-sector learning needs in dysphagia management. It provides valuable insights into both professional and non-professional perspectives and offers actionable recommendations for training development.

I have only some minor comments:

  1. Clarify sampling limitations and discuss potential biases more explicitly.
  2. Strengthen the theoretical framework linking findings to adult/continuing education models.
  3. Streamline tables (e.g., summarize key comparisons visually).
  4. Results could include more interpretation alongside data, particularly on cross-country or demographic patterns.
  5. Shorten and refine the abstract and Discussion for conciseness and clarity.

Author Response

Response to Reviewer 3 Comments:

Thank you very much for taking the time to review this manuscript. Please find the detailed responses below and the corresponding revisions/corrections in track changes in the re-submitted files

Point-by-point response to Comments and Suggestions for Authors:

  • General comment: This is a well-written, structured, and relevant manuscript addressing a clear and underexplored topic: cross-sector learning needs in dysphagia management. It provides valuable insights into both professional and non-professional perspectives and offers actionable recommendations for training development.

Response: We sincerely appreciate the reviewer's thoughtful and encouraging feedback. We are pleased that the relevance and novelty of addressing cross-sector learning needs in dysphagia management has been recognized. We are also grateful for the acknowledgment of the manuscript's structure and clarity, as well as the recognition that our findings provide actionable insights for both professional and non-professional stakeholders. Your positive comments motivate us to continue working in this research area with the aim of developing effective, tailored training interventions that can enhance dysphagia management across different care settings and ultimately improve patient safety and quality of life.

  • Comment 1: Clarify sampling limitations and discuss potential biases more explicitly.

Response 1: We thank the reviewer for this important suggestion. Previously, limitations were briefly mentioned in the Discussion; we have now consolidated and expanded these into a dedicated Limitations and Future Research section at the end of the Discussion. We believe this addition strengthens the manuscript by providing readers with a clearer understanding of the study's scope and boundaries.

  • Comment 2: Strengthen the theoretical framework linking findings to adult/continuing education models.

Response 2: We appreciate this suggestion. We have now added a paragraph in section 4. Discussion acknowledging the importance of grounding future training development in adult learning principles, particularly given the diverse learning needs identified across different stakeholder groups. We believe this addition strengthens the theoretical foundation for our recommendations without deviating from the exploratory nature of our study.

  • Comment 3: Streamline tables (e.g., summarize key comparisons visually).

Response 3: We thank the reviewer for this suggestion. We have revised our tables to enhance visual clarity by using bold formatting to highlight key comparisons and statistically significant differences.  This visual approach improves readability and allows readers to quickly identify the most relevant patterns in the data.

  • Comment 4: Results could include more interpretation alongside data, particularly on cross-country or demographic patterns.

Response 4: We appreciate this suggestion and have made minor modifications to improve the presentation of key patterns in the Results section. However, we have intentionally maintained a clear boundary between objective reporting (Results) and interpretation (Discussion) to preserve methodological rigor. Detailed interpretation of demographic and cross-group patterns is provided in the Discussion section.

  • Comment 5: Shorten and refine the abstract and Discussion for conciseness and clarity.

Response 5: Thank you for pointing this out. We have, according to the comments of all reviewers, revised abstract and discussion sections.  We have revised the Abstract and Discussion to improve conciseness and clarity, removing redundancy and streamlining the presentation of objectives, methods, key findings, and implications for professionals and non-professionals involved in dysphagia care.

Reviewer 4 Report

Comments and Suggestions for Authors

Addressing Learning Needs in Dysphagia Management: Perspectives from Professional and Non-Professional Stakeholders

Dear Authors,

Thanks for submitting this study to the journal of Healthcare. It was a pleasure reading and critiquing the current study. I hope my feedback is helpful for you to enhance the study.  The study assessed the perceived knowledge and learning needs regarding dysphagia management. The topic is highly relevant and the manuscript is well-structured, clearly written, and methodologically transparent. Nevertheless, some weaknesses limit the study’s scientific rigor and generalizability, which should be addressed before publication.

Notes:

Inclusion of five European countries adds cross-cultural value and helps inform broader educational strategies. The study’s integration with an EU project (INDEED) gives it institutional legitimacy and relevance. However, there was unequal representation across countries (i.e., 40% from Spain). I totally understand that the number of participants included from different countries was affected by the sampling method (snowball), but this should be addressed in the discussion section.

The methodology is described in details, the tables and figures (results section) are clrear and concise. The manuscript includes ethical procedures and acknowledges AI assistance (ChatGPT), demonstrating integrity. The discussion effectively connects results to literature.

The development and validation of the questionnaires was well described. However, no information regarding the reliability and validity of the questionnaire. It is not a good idea to refer the readers to the previous studies without explaining the reliability of the measure across the current sample. If the Cronbach’s alpha was <.7, you must address this and justify it.  Also, please add the questionnaire as an appendix for transparency.

The reliance on the self-reported measure without objective assessment and the great  possibility of social desirability must be addressed as well. Also, you may recommend future researchers to use objective measure (i.e., objective knowledge quizzes, KAP scales)

Expand discussion on why differences exist between professions and countries.

Please clarify this statement and provide a reference “This study involved an anonymous survey without collection of personal identifiers or sensitive health data and therefore did not require formal review and approval by a Research Ethics Committee according to the applicable regulations for anonymous, low-risk survey research. ”

Author Response

Response to Reviewer 4 Comments:

Thank you very much for taking the time to review this manuscript. Please find the detailed responses below and the corresponding revisions/corrections in track changes in the re-submitted files

Point-by-point response to Comments and Suggestions for Authors:

  • General comment: Thanks for submitting this study to the journal of Healthcare. It was a pleasure reading and critiquing the current study. I hope my feedback is helpful for you to enhance the study. The study assessed the perceived knowledge and learning needs regarding dysphagia management. The topic is highly relevant and the manuscript is well-structured, clearly written, and methodologically transparent. Nevertheless, some weaknesses limit the study’s scientific rigor and generalizability, which should be addressed before publication.

Response: We sincerely thank the reviewer for their thorough and constructive feedback. The comments have been invaluable in strengthening the manuscript's scientific rigor and clarity. We have carefully addressed all concerns through substantial revisions across all sections, and we believe these changes have significantly enhanced the quality of the work.

  • Comment 1: Inclusion of five European countries adds cross-cultural value and helps inform broader educational strategies. The study’s integration with an EU project (INDEED) gives it institutional legitimacy and relevance. However, there was unequal representation across countries (i.e., 40% from Spain). I totally understand that the number of participants included from different countries was affected by the sampling method (snowball), but this should be addressed in the discussion section.

Response 1: We thank the reviewer for this observation. We agree that the unequal country representation is a limitation of our snowball sampling approach. This has now been explicitly addressed in the Limitations and Future Research section, where we acknowledge that while this method enabled diverse stakeholder recruitment, it resulted in imbalanced country representation. We recommend future studies with more balanced samples to enable robust country-level comparisons.

  • Comment 2: The methodology is described in details, the tables and figures (results section) are clrear and concise. The manuscript includes ethical procedures and acknowledges AI assistance (ChatGPT), demonstrating integrity. The discussion effectively connects results to literature.

Response 2: We sincerely thank the reviewer for their positive feedback and recognition of the methodological clarity, ethical transparency, and the integrity of the manuscript. In response to other reviewers’ valuable suggestions, we have refined several sections to further clarify methodological and discussion aspects. These revisions did not substantially modify the content but contributed to improving the precision, coherence, and overall readability of the manuscript.

  • Comment 3: The development and validation of the questionnaires was well described. However, no information regarding the reliability and validity of the questionnaire. It is not a good idea to refer the readers to the previous studies without explaining the reliability of the measure across the current sample. If the Cronbach’s alpha was <.7, you must address this and justify it. Also, please add the questionnaire as an appendix for transparency.

Response 3: We appreciate the reviewer's suggestion regarding statistical reliability. However, Cronbach's alpha is not appropriate for our questionnaire because we do not intend to measure a unitary construct or calculate a total score. Each question evaluates distinct and independent aspects of self-perceived knowledge levels and learning needs, and they were analyzed separately according to their nature. As recent methodological studies indicate (Edelsbrunner, 2024; Hussey et al., 2025), Cronbach's alpha is only relevant when items are reflective indicators of the same latent construct. Our approach aligns with a formative) model where items do not need to be correlated with each other (Aguirre-Urreta & Rönkkö, 2024)

  • Aguirre-Urreta MI & Rönkkö M. Reconsidering the implications of formative versus reflective measurement model misspecification. Inf Syst J. 2024;34:533–584. https://doi.org/10.1111/isj.12487
  • Edelsbrunner, P.A., Simonsmeier, B.A. & Schneider, M. The Cronbach’s Alpha of Domain-Specific Knowledge Tests Before and After Learning: A Meta-Analysis of Published Studies. Educ Psychol Rev 37, 4 (2025). https://doi.org/10.1007/s10648-024-09982-y
  • Hussey I, Alsalti T, Bosco F, Elson M, Arslan R. An Aberrant Abundance of Cronbach’s Alpha Values at .70. Adv Methods Pract Psychol Sci. 2025;8(1). https://doi.org/10.1177/25152459241287123
  • Comment 4: The reliance on the self-reported measure without objective assessment and the great possibility of social desirability must be addressed as well. Also, you may recommend future researchers to use objective measure (i.e., objective knowledge quizzes, KAP scales)

Response 4: We thank the reviewer for this valuable comment. We have now addressed this limitation explicitly in the Discussion section (Limitations and Future Research), acknowledging that the reliance on self-reported data may introduce recall or social desirability bias and that perceived knowledge may not accurately reflect actual competence. In response to this suggestion, we have also included a recommendation for future studies to combine self-assessment tools with objective measures, such as validated knowledge quizzes and KAP or KAB frameworks, to provide a more accurate evaluation of knowledge and skills.

  • Comment 5: Expand discussion on why differences exist between professions and countries.

Response 5: We appreciate this insightful suggestion. The discussion has been expanded to better interpret the observed differences among professional groups , which may reflect variations in access to training opportunities, professional responsibilities, and country-specific practices. As noted in the Limitations and future research section, the study design and sample distribution did not allow for multivariate or country-level analyses to establish predictors of knowledge. We have now explicitly acknowledged this limitation and emphasized that future research with larger and more homogeneous samples is warranted to explore potential predictors and cross-country differences.

  • Comment 6: Please clarify this statement and provide a reference “This study involved an anonymous survey without collection of personal identifiers or sensitive health data and therefore did not require formal review and approval by a Research Ethics Committee according to the applicable regulations for anonymous, low-risk survey research.

Response 6: We thank the reviewer for raising this point. The journal editor requested clarification on this ethical aspect before peer review, and we provided a detailed justification at that time. Section 2.4 has been revised accordingly to include this clarification and the relevant regulatory framework supporting the exemption from formal ethics review for anonymous, low-risk survey research.

Round 2

Reviewer 1 Report

Comments and Suggestions for Authors

Congratulations and thank you because you incorporate the majority of the suggestions that i proposed in last review. However i believe that the study needs an ethic comité approval. 

Author Response

Response to Reviewer 1 Comments:

Thank you very much for taking the time to review this manuscript again. Please find the detailed responses below

Point-by-point response to Comments and Suggestions for Authors:

  • Comment 1: Congratulations and thank you because you incorporate the majority of the suggestions that i proposed in last review. 

Response 1: Authors sincerely thank the reviewer for the evaluation and positive feedback.

  • Comment 2: However i believe that the study needs an ethic comité approval.

Response 2: Thank you for this comment. As explained in Section 2.4 of the manuscript, ethics committee approval was not required because this study involved anonymous online surveys without personal or sensitive data, in accordance with institutional and European regulations for minimal-risk research. This determination was confirmed by the Data Protection Unit of the University of Zaragoza, which validated compliance with GDPR (as implemented in Spain through Organic Law 3/2018). We believe our approach is consistent with ethical standards for minimal-risk survey research.

Reviewer 2 Report

Comments and Suggestions for Authors

There remain significant critical issues in the work produced, such that its publication cannot be recommended. The title does not clearly convey the setting or the type of study conducted. The objectives are overly complex and not clearly or unequivocally defined. The concept of double reporting is also unclear and missing for trasparency in supllementary files. The internal and external validity of the study remains vague, particularly regarding the development of the questionnaire. Moreover, the study lacks a structured and validated methodology that would allow for meaningful international interpretation. The production of different language versions of the survey is also questionable. Although the STROBE tool is mentioned, it is poorly or only partially applied, particularly in the methodology section. The discussion lacks a genuine interpretation within the framework of clinical and assistive clinical practice. The conclusions are redundant and do not effectively bring the work to a meaningful close. The bibliography is, in many cases (excluding methodological references), outdated—often exceeding the internationally recommended five-year limit—and therefore may not adequately support the scientific validity of the findings. A general and professional native-language review is recommended.

Comments on the Quality of English Language

See comments

Author Response

Response to Reviewer 2 Comments:

We sincerely thank Reviewer 2 for their continued engagement with our manuscript and for taking the time to provide additional feedback.

Point-by-point response to Comments and Suggestions for Authors:

  • Comment 1: There remain significant critical issues in the work produced, such that its publication cannot be recommended.

Response 1: We deeply appreciate the reviewer's commitment to ensuring the highest scientific and methodological standards. However, we respectfully note that the manuscript underwent extensive major revision following the first round of review, addressing all previous concerns raised by Reviewer 2 point by point. These revisions were positively evaluated and accepted as satisfactory by the other three reviewers, who explicitly acknowledged that all concerns had been properly addressed.

While most of the issues raised in this current round were already thoroughly addressed in our previous revision, we have carefully reviewed all comments once again and provide detailed point-by-point responses below. We remain committed to ensuring the clarity, rigor, and consistency of our work.

  • Comment 2: The title does not clearly convey the setting or the type of study conducted.

Response 2:  We appreciate the reviewer’s comment. We have revised the title in Round 1 to improve clarity. The current title "Knowledge Levels and Learning Needs in Dysphagia Management: Perspectives from Professional and Non-Professional Stakeholders in Five European Countries" now clearly indicates:

- The study focus (knowledge and learning needs)

- The study approach (perspectives = self-reported perceptions)

- The population (professional and non-professional stakeholders)

- The setting (five European countries)

The study design (observational, cross-sectional survey) is explicitly stated in the first lines of the abstract, where readers immediately identify it. We believe this approach maintains clarity while avoiding an overly lengthy title (current: 18 words).

  • Comment 3: The objectives are overly complex and not clearly or unequivocally defined.

Response 3: We substantially revised the objectives section in Round 1, and these changes were positively assessed by the other three reviewers, who confirmed that the revised objectives are now clear and well-aligned with the study design. Specifically, following this same comment from Reviewer 2 (previous Comment 3), we replaced "assess" with "explore" to better reflect the exploratory nature of our study.

  • Comment 4: The concept of double reporting is also unclear and missing for trasparency in supllementary files. The internal and external validity of the study remains vague, particularly regarding the development of the questionnaire. Moreover, the study lacks a structured and validated methodology that would allow for meaningful international interpretation. The production of different language versions of the survey is also questionable. Although the STROBE tool is mentioned, it is poorly or only partially applied, particularly in the methodology section.

Response 4: We sincerely thank the reviewer for these methodological considerations. As noted during the first review round, these aspects were thoroughly addressed in our previous major revision. Specifically, we:

- Detailed the questionnaire development process (Section 2.3)

- Clarified translation procedures for multiple language versions

- Added both STROBE and CHERRIES checklists as supplementary material for full methodological transparency

- Included a flow diagram (Figure 1) illustrating participant recruitment and inclusion

These improvements were guided by the reviewer's earlier feedback and were acknowledged as satisfactory by the other three reviewers.

After reviewing other articles published in Healthcare, we have confirmed that our methodological approach meets the standards required for publication in this journal. We note that similar methodological frameworks have been used in published studies, including an Editor's Choice article that we have now cited in our Discussion section (in relation to methodological limitations).

We respectfully believe that our methodology is appropriately structured, transparent, and consistent with international reporting standards for exploratory survey-based research.

  • Comment 5: The discussion lacks a genuine interpretation within the framework of clinical and assistive clinical practice. The conclusions are redundant and do not effectively bring the work to a meaningful close.

Response 5: We sincerely thank the reviewer for these comments. Both the Discussion and Conclusions sections were extensively revised in the prior round following this same feedback from Reviewer 2 (previous Comments 6, 7, and 8). Specifically, we:

- Expanded the discussion to include clinical and assistive practice implications

- Added a dedicated "Limitations and Future Research" section

- Revised the Conclusions to clearly summarize main findings, highlight differences between stakeholder groups, emphasize role-specific educational priorities, and incorporate implications for tailored learning programs

These revisions were positively evaluated by the other three reviewers, who confirmed that the Discussion and Conclusions are now appropriately focused and aligned with the study's objectives. We respectfully believe that both sections now provide meaningful interpretation within the clinical context and effectively close the work.

  • Comment 6: The bibliography is, in many cases (excluding methodological references), outdated—often exceeding the internationally recommended five-year limit—and therefore may not adequately support the scientific validity of the findings.

Response 6: Thank you for pointing out this comment. We carefully reviewed all cited literature and updated several references where appropriate. We have updated 7 references to more recent publications (within the last 5 years) where appropriate and relevant to our topic. Changes made: 

- Updated references: #1 to #5, #7, #9

- Added new references: #65

However, we maintain the remaining references as they represent foundational/methodological work that remains current in the field.

  • Comment 7: A general and professional native-language review is recommended.

Response 7: We thank the reviewer for this comment. The entire manuscript has been reviewed by a professional native English speaker prior to submission, and the language quality was considered satisfactory by the three other reviewers and the editorial office. Therefore, we believe that additional linguistic editing is not necessary at this stage.

Reviewer 4 Report

Comments and Suggestions for Authors

Thanks for the authors for responding to all the comments. I have no further feedback to improve the study. 

Author Response

Response to Reviewer 4 Comments:

Comment: Thanks for the authors for responding to all the comments. I have no further feedback to improve the study.

Response: Thank you very much for taking the time to review this manuscript again. Authors sincerely thank the reviewer for the evaluation and positive feedback.